# Study on the New Role of Civil and Military Air Rescue Nurses in the Italian Context

Francesca Loi [1], Maria Raffaela Lucchetta [1,*], Claudio Mameli [1], Roberta Rosmarino [1], Giulio Oppes [1], Ronald Jaimes Fuentes [2], Ingrid Dallana Avilez Gonzalez [3] and Cesar Ivan Aviles Gonzalez [1]

[1] Department of Medicine and Surgery, Nursing Faculty, University of Cagliari, 09124 Cagliari, Italy
[2] Clinica Pediatrica Simon Bolivar and Clinica Cesar, Valledupar 2000005, Colombia
[3] Imagen Radiologica Diagnostica S.A.S., Valledupar 2000005, Colombia
* Correspondence: mariaraffaela.lucchetta@aslcagliari.it

**Abstract: Context**: An emergency medical nurse is a health professional who operates at a very high level in the field of emergencies. The nurses of the critical area departments of the Territorial Emergency Department currently operate in the Sardinian helicopter rescue service. The effectiveness of the treatment that these nurses administer can be attributed to the quality of the previous and recurrent training that this unit must undergo. **Objective:** This study's aim was to investigate the role of civil and military helicopter nurses in the context of Italian medical aid. **Methods:** A qualitative study, with a phenomenological approach, was conducted by interviewing 15 emergency medical nurses, using detailed recordings and transcripts. These findings were then compared to understand how nurses work outside their department of origin, how their training has influenced their ability to establish themselves outside it, and thus their ability to become part of a context considered to be of the highest level. **Participants and research context:** The personnel interviewed in this study were those who were working in the helibases of Cagliari, Olbia, and Alghero. The limitations of this study are linked to the impossibility of obtaining an internship at a company, because, at the time of the study, an agreement between the university and the Areus company was not active. **Ethical considerations:** Participation in this research was completely voluntary. In fact, the participants could cease participating at any time. **Results:** This study revealed issues related to training, preparation, motivation to carry out the role held, nursing autonomy, the willingness to collaborate between the various rescue organizations, the use of the helicopter rescue service, and possible improvements for this service. **Conclusions:** civil air rescue nurses can deepen their knowledge by examining the work of military air rescue nurses, because, although the operational contexts are different, some techniques used in a hostile environment are also applicable to civilian environments. By doing so, nurses could become independent team leaders for all intents and purposes, managing their own training, preparation, and technical skills.

**Keywords:** emergency department; military helicopter nurse; civil helicopter nurse

## 1. Introduction

Helicopter rescue is a rescue activity that is carried out with the use of dedicated helicopters and guarantees high-level assistance with rapid intervention times, and therefore better results, especially with respect to health interventions, compared to those carried out by road. Currently, the Italian helicopter rescue service is organized on a regional or provincial basis and has been included in the context of the emergency telephone numbers "118 or 112" (i.e., the numbers assigned to the health emergency service). Helicopter rescues provide first aid and emergency medical transport to the most suitable site, for the advanced assistance of the patient. With the aim of guaranteeing, managing, and homogenizing rescue services in the territory of the region, the regional emergency and urgency of Sardinia (AREUS) was established under a regional law (17 November 2014, n. 23). This

agency is responsible for coordinating tasks related to emergency/urgent care services currently carried out by 118 operations centers of different healthcare companies, including: helicopter rescue service, transport of people (including newborns), organs, and tissues; exchange and compensation of blood and blood components; and ensuring coordination with all the companies of the regional health system and the institutions involved, in order to guarantee the effectiveness of emergency and urgent healthcare responses [1]. This agency is also responsible for the medical equipment and medicines supplied for the execution of advanced resuscitation maneuvers and the treatment of traumatized people. The HEMS (Helicopter Emergency Medical Service) allows for emergency medical interventions to be performed directly at the scene of the event, even with off-field landing, in order to reach a patient and facilitate their recovery. Additionally, this service facilitates special operations, such as the loading of a stretcher with a winch. Moreover, SAR (Search and Rescue) interventions can be employed for rescue interventions in an impervious environment, while air ambulance interventions can be used for the movement of patients from one hospital to another. In war contexts, search-and-rescue interventions are performed by the Italian armed forces, who, unlike the HEMS that operates in a non-war context, have lower safety standards, because their aim is exclusively the patient recovery, not the on-site treatment, which is performed in the shortest possible time to reduce the exposure of rescuers to hostile fire. For this reason, the training that civilian and military personnel receive before they can be deployed in search-and-rescue operations differ due to the contexts in which they must operate. Therefore, it is necessary that professionals in this sector, such as helicopter rescue nurses, who have greater knowledge pertaining to the training of non-commissioned health officers, are able to operate in an interchangeable way that incorporates the necessary knowledge included in both types of rescue missions, because no two interventions can ever be the same. It is thus imperative to shape the knowledge possessed at each individual intervention. The discussion of interventions that take place in an urban environment encourages the current theory and the right degree of experience in the field leading to a successful operation. However, when the recovery operation is conducted in an impervious environment, improvision is often required, and it is difficult to always operate with complete safety. For this reason, when conducting this study, we did not want to limit its scope solely to the civil environment.

Thus, by incorporating procedures from the military field, we can adopt the appropriate notions that would adequately account for increasing the knowledge of civilian health workers engaged in helicopter rescue.

Purpose of the Study

The main objective of this study was to explore the operational context of emergency medical nurses and understand how they work outside their departments of origin, how their training has affected their ability to establish themselves outside it, and thus how they were able to become part of a context that is considered of the highest level. Furthermore, we sought to determine the Sardinian air rescue nurses' familiarity with the techniques used in the military, and their willingness to develop their cultural backgrounds and opinions regarding the current regional air rescue service in a new working context, which could also provide new degrees of personal satisfaction. The previous levels of personal training of each study participant were also investigated, alongside their willingness to increase their professional culture knowledge.

In addition to the theoretical part, and therefore purely to health training, specific training is also necessary from a technical point of view, in order to be able to approach inaccessible environments without endangerment and being able to operate at an optimum.

## 2. Materials and Methods

This study was conducted using an interpretative phenomenological analysis, which is a qualitative research approach that values 'a detailed experiential account of the person's involvement in the context' [2]. Interpretative phenomenological analysis allows for catching communication meanings through the narration of participants' experiences within a cultural, social, and personal perspective, hence it can be said that the method implies an interpretative approach, enriched by descriptive notes [3]. Following interpretative phenomenological analysis, philosophical roots that come from Heidegger's philosophy, and meanings are always created through interactions, including those with researchers [4]. According to this, the researchers' preconceived concepts and personal attitude cannot be kept apart from the investigation, but they can represent a tool to conduct the analysis [5]. On this basis, interpretative phenomenological analysis develops a double hermeneutic circle, where 'the participants are trying to make sense of their world; the researcher is trying to make sense of the participants trying to make sense of their world' [6]. According to the idiographic focus of interpretative phenomenological analysis, this study explores the perspectives of individuals in their unique context of life. Alongside that, this study adheres to interpretative phenomenological analysis request to illustrate and describe themes by a rich reporting of excerpts from participants' accounts, taking into account the 32 criteria of the qualitative research for semi-structured interviews [7]. We carried out the study with interviews conducted by FL, a female student of the nursing degree course, with previous work experience in the emergency/urgency department, because she has been a part of medical aid association for 7 years. The relationship with the interviewees began on the same day as the interviews, as the researcher did not know the interviewees, but they were informed during the beginning of the interview about their personal goals. There was also a personal motivation for conducting the aforementioned study. The method encompassed a phenomenological orientation and in order to implement this, the 40 nurses operating in the Sardinian regional helicopter rescue service were selected, 15 of whom decided to join the project completely freely, thus becoming the champions of the study. Those who decided not to participate did so due to a lack of time. Participants were invited via email and the interviews were conducted via video call. During the interviews, no other people outside the study were present. The participants all came from critical area departments or from the local emergency service, and were mostly male. All respondents had considerable previous work experience. The questions have been designed to be compatible with the chosen topic, in full respect to ethics and privacy. Questions selected for the interviews were relevant to the research, focusing on a topic with brevity, clarity, and appropriate language for the respondents. Furthermore, the delicate topics were not addressed directly, so as not to make the participants uncomfortable, and above all, no double-answered questions were formulated, and space–time contexts were delimited. The pilot interview was submitted to two helicopter rescue nurses operating in the Sardinian region. They did not suggest any changes to the questions, and they were clear and concise. This was structured in a way to understand how the respondents spoke of the investigated phenomenon, allowing them to respond with a natural language, and thus have immediate answers.

All the interviewees were asked the same questions and data collection was carried out by recording the interviews, in order to have the necessary material to work satisfactorily on the study and to avoid errors. The total duration of the interviews was s 10.40 h, after which the saturation of the data obtained, with respect to the questions, was evaluated. Having reached the set target, the data was encoded through revision by FL, MRL, CM, and CIAG, with the drafting of a coding tree and the derivation of the themes that emerged from the data obtained with the interviews, identifying the main thematic areas and extrapolating the answers of each participant in the interview for each question proposed. Not all participants wanted to review their interviews. No data-processing software was used. At the end of the interviews, the participants provided positive feedback for their involvement in the project and for the chosen topic, with the hope of a positive evolution of the project. To ensure the anonymity of the participants, identification numbers were assigned, i.e., each

participant corresponded to a number which was used for reporting during the discussion of the data and the text sentences said during the interviews, allowing for an efficient comparison with the data obtained from the literature review.

With the analysis of the results obtained in the interviews, coherence emerged between the authors who analyzed the information, the data presented and the results obtained. Above all, with the results, the themes assigned in the study project clearly asserted themselves. Moreover, some issues of secondary importance and not foreseen in the project emerged, which were discussed during the interviews to also gain a personal opinion from the interviewees.

## 3. Results

### 3.1. Data Collection

To proceed with the investigation, 15 emergency medical nurses from the Cagliari, Alghero, and Olbia bases were interviewed. In addition, with the total number of interviews, information saturation was obtained. This made it possible to obtain fairly comprehensive information, and thus be able to successfully conclude the study. Ten questions were administered, whereby the interviewee had full freedom of contextual and verbal expression.

The questions administered during the interviews were as follows:

1. Can you tell me how the idea of becoming a heli-rescuer nurse came about and what health context/hospital ward it comes from?
2. What can be done concretely to increase the level of knowledge of the techniques? How could field training improve with periodic refresher courses and how could it affect your performance at the event?
3. What were the first interventions you made and how did you find yourself managing them?
4. How much would you be willing to increase the training received and what would you change?
5. What do you think of a possible periodic training being done in collaboration with other rescue units, such as basic ambulances or MSA, and the drafting of updated guidelines for the cooperation between the different rescue bodies?
6. Do you know the differences between military and civilian evacuation techniques?
7. Currently, the healthcare team leader is represented by the figure of a doctor, but in the future, the nurses could be given more autonomy in operations. How do you think this can happen? What do you know and what do you think about nursing see-and-treat, and how could this shape future education?
8. How does whether or not being a nurse on board a helicopter make a difference?
9. What do you think of the current regulation of air rescue and the uses made of this service, and what do you think of the management of events by the operations centers?
10. What do you think can be done to implement the current HEMS?

The interviews were recorded and transcribed in detail, and then the different opinions of each study participant were compared.

### 3.2. Inclusion and Exclusion Criteria

3.2.1. Inclusion Criteria

Serve as a nurse in the Sardinian regional helicopter rescue service.
Willingness to participate in the study through an interview.

3.2.2. Exclusion Criteria

Do not operate at the HEMS of the Sardinian region.

### 3.3. Ethical Implications

Interviews were authorized by AREUS and all the participating nurses.

Collaboration on the project took place entirely voluntarily. The subjects involved also had the possibility to interrupt their participation at any time. The anonymity of the data was guaranteed, using encrypted names for the identification of the participants. Finally,

the nurses were made aware of the data and results obtained from the analysis, in order to be able to express a final approval, and thus, the publication of this paper could proceed.

*3.4. Participant Characteristics*

Fifteen emergency medical nurses from the Cagliari, Alghero, and Olbia helibases, male and female, from different operational departments, but always in the context of the health emergency, were interviewed (Table 1).

**Table 1.** Home base, department of origin and gender of participants.

| Home Base | Department of Origin | Gender |
|---|---|---|
| Cagliari-Elmas | First aid, 2<br>Anesthesia–Resuscitation, 4<br>Territorial service or operations center, 4 | Males, 8<br>Females, 2 |
| Alghero-Fertilia | Emergency room<br>Anesthesia–Resuscitation, 1<br>Territorial service or operations center, 1 | Males, 2<br>Females, 0 |
| Olbia | First aid, 1<br>Anesthesia–Resuscitation, 1<br>Territorial service or operations center, 1 | Males, 2<br>Females, 1 |

From the data collection, which are represented by the semi-structured interviews addressed to the nurses of the HEMS of the Sardinian region, it was possible to identify different thematic categories:

1. Context of origin
2. Training, preparation, and factors influencing training and professional growth
3. First interventions and management
4. Increase in training and/or changes to be implemented
5. Collaboration and new guidelines with other rescue units
6. Knowledge of differences between civilian and military evacuation techniques
7. Nurse team leader: the nursing see-and-treat
8. A nurse on board the air ambulance can make a difference
9. Regulation/management of interventions by the operations center
10. Implementation of the HEMS

The following considerations are based on the analysis of the interviews:

*3.5. Context of Origin*

The nurses interviewed all come from critical area departments, such as the emergency room or anesthesia and resuscitation service, or from urgent emergency contexts, such as 118 or operations centers.

3.5.1. Training, Preparation, and Factors Influencing Training and Professional Growth

All the nurses interviewed came either from a critical area department or from a territorial emergency service, and therefore had training that coincides with that of an emergency medical nurse, since they work in emergency situations in the management of critical patients. The critical area nurses, working in this context, can not only put their acquired knowledge and training to good use, but must also recognize and control their reactions towards a particular, and in some ways dangerous, environment which is that of the helicopter. The initial impact of this unusual environment was different for the different participants in the interviews, but overall, no emotions were shown that could have affected operational performance. Almost all the nurses interviewed approached the helicopter for the first time in 2017/18, when the helicopter rescue service of the Sardinian region was opened, a service that was previously performed by the Armed Forces and Fire

Brigade. Some participants, such as the no. 2, no. 4, and no. 9, had already served during past collaborations with the Fire Brigade, Air Force, Italian Army, and Coast Guard.

The periodic courses were essential for maintaining and increasing knowledge and for the best possible performance during rescue operations, but due to various organizational and economic factors of the training liquidator company, unfortunately they were not provided with the right timing, but rather with very long delays.

In this regard, nurse no. 1 stated that 'The human resources team should have more sensitivity with respect to the training and heritage of each operator, because this affects the training and work of workers, because everyone is trained to do this that belongs to him and so he can do it better'.

No. 2 added, 'From a health point of view, training is continuous and is linked to the expiry of the courses already taken. Unfortunately, the training capacity is slow and fragmented because the courses have a limited number of participants, and it would therefore be desirable for the whole helicopter rescue team to take the courses at the same time in order to have all the personnel on the same level'.

No. 3 focused on the completeness of the courses and stated, 'If the training path is complete, the employee does not need to spend time with a self-financed course since I only go for a self-financed course if I feel that I have gaps on that particular topic but in a system that works, my skills and competences should be monitored and in case of gaps it should be filled immediately. Training is within the profession and as a professional I have a total relationship with my employer and in turn the company must give me all the tools to be so'.

### 3.5.2. First Interventions and Management

Most of the nurses interviewed stated that during the first interventions, the main feeling was adrenaline and strong emotion. Most interventions were related to trauma and only a few to time-dependent pathologies, such as cardiac arrest or airway management.

No. 2 stated, 'Among the first interventions I remember a trauma at home. Initially there was a little excitement for the new work environment but then, once a relationship of trust was established with the rest of the team, the difficulties diminished and there has never been anything that has affected the operations in event. By doing the helicopter rescue service it is possible to see the pre- hospitalization part and allows to implement the prior knowledge in another part of the assistance'.

No. 4 added, 'In the first interventions he found himself somewhat conditioned by the machine with which he found himself working, the reduced space opens up and due to the fact that he was no longer on the wheel the principle with which one must always operate the safety and on board a helicopter changes a lot compared to an ambulance because the procedures are different and must be respected in order not to compromise safety'.

For no.6, the first interventions were a major facial trauma and cardiac arrest on the beach. 'The sensation was not uncomfortable, because I felt at ease doing mountain rescue, also thanks to my physical prowess and my habit of working to recover injured patients. So, it wasn't something that had a big impact but definitely done at a higher level'.

No. 13 recounted 'The first intervention was an electric shock. Compared to the ward where you work in a "protected" environment, you are there alone with the doctor and with the material available. From a nursing point of view, it has been rewarding and educational and what a nurse can learn from such an experience, marks for a lifetime. It is and must always be a give and take with one's colleagues and thus one can have good professional autonomy and one can derive gratification from this'.

### 3.5.3. Training Increase and/or Changes

Among the nurses interviewed, it was commonly thought that with regard to the increase in training or changes that could be implemented, it is necessary to improve the cadence in the provision of courses.

No. 4 stated in this regard, 'The training courses should be better scheduled in order to have constant retraining because lately due to the covid the updates have been slowed down or even suspended. They are fundamental'.

No. 5 added, 'I would improve the frequency of courses not only for helicopter rescue but also for out-of-hospital rescuers. The basis for an optimal rescue chain is that we all speak the same language and therefore move with the same protocols. I would like to interface with the stations, with the guys from the emergency vehicles, I would like to have a relationship between people who already know each other and already know how to work together. In the last emergency course, we had the chance to meet MIKE and INDIA rescuers; therefore, we are moving towards this reality. So as training I wouldn't change much'.

For no. 15, 'The thing to change would have been the timing of the training, because the training was very fast and compact. The type of training is excellent, but it should be implemented with more technical and not just theoretical training such as, for example, with regard to the crash in the cab, flooding tests in the swimming pool or impervious environment when the machine has technical problems and there is the need to evacuate quickly.

As for no. 13, they concentrated their opinion on the improvement of the technical part and stated, 'I would deal more with the technical aspect to fill some gaps than fundamentally from the theoretical point of view, as far as the health one is concerned, I feel I have'.

### 3.5.4. Collaboration and New Guidelines with Other Rescue Units

The nurses interviewed expressed a common opinion on being in favor of training with other rescue agencies, such as basic ambulances or MSA, as this would be a further element of improvement of the current working conditions of the helicopter rescuer.

Nurse no. 1 stated that it would be desirable to draft unified protocols for cooperation from various emergency bodies in order to operate at their best and always in safety, because protocols are fundamental in any environment to optimize times and work, and following a common ABC to have the best performance for the good of the patient.

Nurse no. 2 added, 'The drafting of any protocols would be desirable, because it often happens to work with staff who are not able to operate safely in the event of a rendezvous with the helicopter, and thus you compromise the security of the event in which you are supposed to operate. Basic training would be required for all personnel with meetings and simulations of intervention to learn how to best interface with the helicopter operators to avoid obstacles'.

Among the interviewees, there was a recurring desire for cooperation between the rescue agencies, in order to operate in the best possible way and in synergy in any situation. The absence of guidelines emerged to standardize the operations between them in the event of the involvement of different agencies, as noted by nurses no. 1 and no. 5, while no. 12 also offered training and work exchanges in order to allow all nursing staff to rotate in all departments, to have a complete 360° training.

### 3.5.5. Knowledge of Differences between Civilian and Military Evacuation Techniques

Of the nurses interviewed, only four nurses out of 15 were aware of the differences between military and civilian evacuation techniques, and this was only because there was an increase in knowledge on a personal basis and only to broaden one's cultural background.

No. 4 knew of its existence, 'Because the scenarios are different and therefore the approaches also change, the substantial difference lies in the fact that in the military environment the rapid evacuation of the wounded is essential, thus quickly abandoning the hostile intervention area and therefore operating in unsafe areas in a purely scoop-and-run style, while as regards the civilian scenario if the scenario should not be safe, an attempt is made to make it safe through the bodies in charge in order to be able to operate on site as

well to stabilize the patient before evacuating him because operating inside the helicopter is difficult due to the limited space'.

No. 5 stated, 'I was lucky enough to take the course on maxi-emergencies. The management of more than three injuries becomes really difficult to manage, you have to activate mechanisms other than the standard. In these cases, the firefighters and the police forces enter the scene and here collaboration becomes fundamental. The recovery teams move in organization with the plants. For the evacuation of military forces, probably, a lot is similar to ours. When I worked for the Coast Guard, I saw that the scoop-and-run used is the one we use ourselves. We try to apply it as a weapon to speed up the intervention'.

No. 6 said, 'I know them because I took the qualifying course with the CRI called amalgam which was born for the training of health personnel to be sent to a war context and then out of a desire to increase one's knowledge. At a military level, triage and rescue are done not to the most serious z, but to the one who has the most possibility of continuing to fight and therefore evacuating it as soon as possible with the care under fire technique, even at the cost of sacrificing some rescue techniques which instead it would have applied in a civilian scenario. Another difference is that the least serious one embarks first and the most serious last in order to be able to unload it first. In triage of maxi-civil emergency it is unlikely that there are armed threats. In the civil context, more importance is given to the most serious but also to the salvageable'.

No. 13 said, 'I did some courses concerning the medevac and I've also worked with the air force and so I've been able to see the differences in the approaches to the wounded and the different scenarios, but always everything in broad strokes'.

### 3.5.6. Nurse Team Leader: The Nursing See-and-Treat

Despite the continuous evolution of nursing into a real profession, and no longer mere executors of medical orders, the idea of the team leader doctor remains among many nurses, except in conditions in which one must take care of more patients, and therefore it "divides the roles". The nursing sees-and-treat is slowly gaining more and more ground in healthcare realities and, with the acceptance of this, the nursing evolution will certainly undergo a big acceleration. In this regard, nurse no. 10 declared that the nurses must certainly have courage, confidence, awareness, and think like a team leader in order to be able to cover this role.

No. 1 pointed out that 'The see-and-treat protocol is a correct practice only if there are shared protocols, but that, despite this, it still remains a double-edged sword, because the border of abuse of the medical profession is around the corner. Currently in some Italian regions these protocols are already a reality but only because they have been drawn up, shared and signed by doctors and nurses'.

No. 6 confirmed the protocols and stated, 'In some situations the nurse is already a team leader and this has also had excellent results. What helps is having operational protocols to avoid invading medical operations and vice versa, finding solutions to avoid making a medical diagnosis, but also contacting and involving the doctor by telephone who can, thus, prescribe any therapies and all on a registered line for the protection of operators'.

For No. 7, 'Currently the doctor is the team leader, but an AREUS project is underway for the "strengthening" of the figure of the nurse through a selective and qualifying training course with advanced courses, including time-dependent pathologies with shared protocols with CO doctors and Mike ambulances. It is a historic moment for the Sardinia region, because in other regions, this is already a reality and here everything is slower. The see-and-treat in the emergency rooms is taking place and, however, better training of the personnel in service is foreseen and a weaning from the figure of the doctor is necessary. However, it is essential to be careful to keep the figures of doctor and nurse distinct so as not to fall into what could be the "mini doctor" or "paramedic"'.

No. 8 added, 'There is a continuous search for nursing autonomy, but, to tell the truth, there is a dichotomous theory: without medical culture, there can be no nursing culture. They are two things that go hand in hand and must go hand in hand. More than a see and

treat, it would be more correct to speak of know see-and-treat, because having the basics one can treat and "dare" knowing the limits to guarantee the patient the best possible treatment. We must always act in science and conscience'.

### 3.5.7. Nurses on Board the Air Ambulance Can Make the Difference

The idea that the nurses on board a helicopter can make a difference was a common thought among the nurses interviewed, as both the legacy and the professionalism exercised affirm this theory. For no.10, 'The nurse makes the difference because he is not only an operator who performs simple procedures for the patient, but collaborates with a clinical eye to give the best for the patient and therefore his presence is essential because he is also equipped with excellent communication skills and can thus communicate with the operations center or with other elements on board while the doctor operates on the patient and is aware, alert and careful about what and how to communicate because he is aware of what the doctor is going to do, being also health professional'.

According to no. 1, 'Having a nurse on board makes all the difference, as without one, it would be a scoop-and-run. The nurse and the doctor are able to complete the picture of patient management. In environments such as high mountains, the nurse is often not present on board only for reasons related to the weight of the aircraft and/or limited spaces. The nurse collaborates with the doctor to obtain the best to provide to the patient and if his figure is missing, there is the risk of compromising the success of the intervention because the nurse is irreplaceable with other figures. What a helicopter does is bring resuscitation over the territory and as within the hospital walls, the doctor/nurse ratio must be 1/1. The doctor has many roles and to better distribute them the figure of the nurse is necessary and in doing so it is possible to optimize times'.

For no. 8, 'In the civilian field, there are five figures on board: pilot, on-board technician/specialist, helicopter rescue technician, doctor, nurse who can be replaced by the figure of a dog lover. In order to be fundamental on board, the nurse needs to become a key element and therefore cannot aspire to the importance of the other figures. It is a point of view that diverges from that of many colleagues. The nurse should have more autonomy, but due to a medical cultural deficit, the doctor will always prevail over this figure'.

According to no. 15, 'If the nurse is missing on board, there is certainly a missed possibility of providing a complete health service, because otherwise, it would be a single recovery of the person without a health treatment. Interventions in inaccessible places and on the territory always require a sanitary action offered by a doctor and nurse, so much so that in the high mountains, where there is only a doctor on board for logistical reasons, the intervention is only a recovery'.

### 3.5.8. Regulation/Management of Interventions by the Operations Center

With regard to the current regulation of the air ambulance and its uses:

No. 3 stated that 'Does the current regulation need ideas for improvement which for example can be the "rules of engagement", is when the hospital needs to activate the helicopter rather than an MSA? Because it is not a mathematical algorithm where above a given level there is automatic activation of the ambulance or below this value the ambulance call is mandatory, but there is an interpretative area in which one must get to understand what the better tool for the needs of that moment and reducing this to a simple "yes" or "no" is difficult to narrow down. What certainly can and must be improved are the activation times, also because if one thinks superficially, the helicopter could be suitable for everything, it can reach everywhere, but one must always reflect on the timing and therefore, for each intervention one must evaluate all the variables in a short time'.

For no. 1, 'The operations centers carry out their work at their best but within the limits of their capabilities: there are still communication gaps between the COs and the means on the territory such as the right coordinates of the target, identifying the right means to send that always is the closest but the fastest. It is necessary to improve communication in order to be able to better define the needs of both parties. It would be necessary to implement the

briefings with the discussion of the cases that have just been carried out and to understand what to improve, not only among the helicopter rescue team, but also with the CO'.

Nurse no. 10 stated that 'The helicopter rescue in Sardinia was the icing on the cake to complete the regional rescue sector and therefore, being a new service, it needed time to consolidate and be managed in the best possible way and therefore enter into everyday life'.

According to no. 13, 'In Sardinia, there are two operations centers and they operate in different ways as regards the activation of the helicopter rescue: the CO of Sassari has a legacy from when they worked with DRAGO, the fire brigade helicopter and has therefore maintained that mentality because uses the helicopter not as a means of rescue, but as a means to be used in case of extreme necessity. The CO of Cagliari, on the other hand, makes better use of it, as a resource to be used in the best possible way'.

No. 15 stated that 'The desirable thing would be to create a single operating entrance and in fact this project had already been proposed. The current problem is the lack of communication between the two control panels and therefore this translates into a problem in the management of events precisely due to a gap in communication. This brings up timing issues in patient centralization'.

### 3.5.9. Implementation of the HEMS

The current HEMS is efficient, but can be implemented with various expedients, proposed by the nursing staff operating on board the helicopter rescue service, including increasing the number of helicopter bases, relocating them, upgrading the night service and improving communication between operations centers and rescue units.

According to no. 5, 'We are a big family, so I know all my colleagues and I also try to interface with events that have already taken place. So, it might be a good idea to hold meetings with colleagues to understand what the critical points are. Once a month, therefore, meetings could be held with the coordinator and with the colleague, these are things that make a team and a team. To grow you need to confront and communicate with all those who participate in the service'.

The common opinion among almost all the nurses interviewed was the need to expand the service to also cover the night hours after the ephemeris.

No. 7 stated in this regard, 'To implement the service, it would certainly be necessary to increase the 24/7 operations of the Cagliari base and create real night operations, with the possibility of primary interventions even at night'.

An idea of relocating the helibases to better balance the service also emerged.

No. 12 said, 'The helicopter bases should certainly be better located and it would be right to increase operations also with regard to the night service because currently only Olbia operates as a night base. Currently the location is unbalanced, because there are two bases in the north and only one in the south, and therefore it would be excellent to move or add one to the center, in this way there would be homogeneity and certainly optimization of the service with reduction of times especially for issues related to logistics. It would also be a good implementation to move the AW 139 helicopter to the center because it is the fastest and most efficient'.

An idea was also shared by others, including no. 10, who stated, 'Surely the network of helicopter rescue bases should be improved and distributed better. Furthermore, it is necessary to continue with the training of all on-board operators in order to have an increasingly efficient service'.

## 4. Study Limits

The limits of the study are linked to the non-possibility of internship in the helicopter bases, since an agreement between the University of Cagliari and the AREUS company was not active at the time of the study, as a study carried out at 360° would probably have been more complete. Due to its social and organizational characteristics and as a qualitative study, the results obtained cannot be extrapolated to other contexts. In addition, one of the authors of this study works as a nurse in this institution, and this can influence the responses and behavior of the interviewees.

## 5. Discussion

The main purpose of the study was to define the effective activity and interventions that characterize the figure of both civil and military air rescue nurses, and collect their points of view regarding the role covered, their responsibilities and their own experiences, and possible proposals for improving the service. It appeared that currently, there are no operational protocols for the region of Sardinia concerning the cooperation between the various rescue bodies in the event of maxi-emergencies and the approach to the helicopter. It also appeared that the cadence of the training courses was not deferred in an appropriate manner for the effective maintenance of the knowledge acquired with the previous courses, a point with which Imbriaco [8] and Marinangeli also agree [9]. Rather, it would be necessary to organize them more often and, above all, that they have the enabling function and not just training. The result is that communication between different entities and the HEMS often has gaps deriving from not operating with unified protocols, and therefore speaking a common language and also breaking down language barriers, as Hatzfeld dealt with in his study [10]. In the study, he talks about the communication barriers in pain management, where he defines that the development of solutions to address these factors is necessary, because it should be a priority to ensure that pain is adequately managed during transport.

It emerged from the interviews that most of the nurses during the first interventions felt adrenaline and strong emotions for the new operating context they were facing, because the approach to rescue with the new means subjected them to stressful conditions, and this fact was confirmed by studies by Carchietti [11].

Civilian health training aims to operate safely in any condition, while in the military context, the aim is to rescue injured personnel in a hostile environment, working in any condition, safely or not, and with ad hoc protocols based on studies conducted in the past war events, such as those made by Ternus [12]. In the civil context of maxi-emergencies, more importance is given to the most serious, but salvageable, injured patients, in order not to incur the loss of precious time, which instead, perhaps, would have allowed the saving of several lives involved in a given event. In the civilian context, if more people have to be embarked in an air ambulance, the most serious is loaded first in order to be able to immediately provide the necessary treatment, and the least serious last, a procedure opposite to the military context, where the most serious is loaded last to preserve the safety of moderately injured personnel and be able to allow the most seriously injured to be the first to receive the necessary treatment once landed in a safe area, as ascertained by Staudt's studies [13].

Another point that emerged is that the nurse could, in a future scenario, be a team leader for all intents and purposes, without obviously invading medical operations, finding solutions to avoid stumbling upon a possible medical diagnosis, but collaborating with this figure, so that there is the protection of both operators; (J K Maddry [14]). In fact, in his study, he found that it was the nurse who provided the greatest care in terms of administering drugs and treatments, and it is also consistent with the fact that by combining their skills and knowledge, the nurse and the doctor manage to work in synergy and give the best of themselves in favor of the patient.

What has emerged, both from the interviews and from the studies conducted by Christensen [15] and Sutherland [16], is that military evacuation techniques involve operat-

ing in unsafe conditions, since one finds oneself evacuating personnel under enemy fire often in a hostile environment, and therefore the patient is not stabilized on the recovery site, as is often done in civilian life, but only when the advanced medical fields arrive, and therefore the treatments are also postponed. Furthermore, due to logistic issues, there is rarely a nurse in military helicopters. One point on which the HEMS could improve would be active listening to the staff proposals by the logistic service, and also aiming to improve the supplies of on-board equipment, thus optimizing expenses. This is because one must not only think in theoretical terms, and therefore how good that device can be, but also in practical terms to test whether or not a given device can actually be useful in that scenario/context. The region of Sardinia, in collaboration with the HEMS, should design a better location for the helibases and helipads, in order to increase the level of operations and improve the service, also activating a night service which is currently only carried out for secondary interventions, such as intra-hospital transport. With his study, Christensen [15] dealt with the issue of the importance of the time factor in the administration of care, and also ascertained that with the helicopter, it is possible to reduce patient transport times. Carchietti also agrees on the same issue [11]. Regarding the technical training of air-transported personnel, also for Imbriaco [8], it would be advisable to increase physical training and aim at autonomous work, even without the constant support of the helicopter rescue technician for any maneuver in an inaccessible environment, and thus release them from the heavy responsibilities of the movement of personnel and materials, even if the idea of teamwork remains essential.

## 6. Conclusions

The main purpose of the study was to define the effective activities and interventions that characterize the figure of both civil and military air rescue nurses, and collect their points of view regarding the role covered, their responsibilities and their own experiences, points of meeting and possible proposals for improving the service.

The main objective of every health worker operating in an emergency is to 'give' time to the patient, so that they can reach a hospital from the place of injury, where the staff can provide the correct application of the operating protocols, to save their life, limb or function. The 'purpose' of the work, whether by a civilian or military nurse, is to make a difference. Thanks to the evolution of intra- and extra-hospital emergency nursing, in the near future, it will be possible to guarantee the patient greater chances of recovery and a better quality of care. To make this happen, there must be a constant and growing search for the improvement of one's cultural resources, and a commitment to allow nursing to evolve more and more. Cooperation and communication between health professionals are points to improve in order to obtain a service that is already a flagship, but which still has considerable room for improvement. Some Territorial Emergency Systems use methods of approach to maxi-emergencies imported from foreign realities without verifying their applicability on our territory, and to date, we do not know how many 118 systems are equipped with a plan to deal with maxi-emergencies. The statement can also be extended to hospitals, despite the fact that training on the subject has been more widespread [16]. It is necessary to encourage and support research in this area and the establishment of extra courses, even during university studies, in order to be able to expand this environment, and increase and improve both the skills of the individual and of the entire profession.

Hand in hand with professional growth, numerous achievements can also be highlighted for the patient, such as having at least one more chance of survival thanks to the timeliness and quality of the care provided. The helicopter rescue is still a very young service, and therefore with ample room for improvement, and this will be possible if all the figures involved in this service actually cooperate in unison to ensure the best possible outcome of the ideas still in the pipeline. The emergency medical nurse is certainly the professional figure with the greatest potential for growth in this sector, since, by gaining more and more autonomy, she could become a key figure for the team in the near future. The higher risk ratio for HEMS missions, when compared to ground rescue, requires a

rigorous quality management system. When it comes to missions in remote and exposed areas, the scope of medical treatment must be adjusted to the individual situation. Medical competence is key in order to balance guideline compliance, or maximal care versus optimal care characterized as a mission-specific, individualized emergency care concept. Nevertheless, medical decision-making and treatment is typically based on the best scientific evidence, personal skills, competence, and the mission scenario will determine the scope of interventions suitable to improve outcomes. Thus, the profile of requirements for the HEMS medical crew is high [17].

With continuous training and new discoveries in the fields of medicine and nursing, this service, which already aims at excellence, can certainly be used in a concrete way to the fullest of its abilities. Furthermore, by creating unified protocols for cooperation between various entities operating in emergency situations, it will be possible to have better performances, and not only will the working conditions of the helicopter rescuer improve, but also the success of the interventions, and therefore the well-being of the patient. It is essential to operate through shared protocols, because in this way, the nurse can avoid any encroachment on what the medical profession is, but still operate independently and to the best of their abilities. However, the figure of the nurse is very important and if their figure were to be missing, there would be the risk of compromising the success of an intervention, because a helicopter used in rescue brings resuscitation to the area, and therefore, as it should being within the walls of a hospital, it is necessary to divide the tasks between doctors and nurses, in order to obtain effective cooperation and an optimization of roles and intervention times. The region of Sardinia is unfortunately penalized with regard to the orography of the territory, and therefore the regional helicopter rescue service can make the difference in terms of lives saved, as it remains a valid and more efficient alternative to road rescue. By implementing the service and increasing the education and training of air-transported personnel, with both the technical and health operators carrying out the much-needed 24 h service, the Sardinian air rescue service will be able to be a real flower to the buttonhole, and thus make the difference by reducing mission times for the transport of patients in need of treatment to the most suitable structures and with the provision of an excellent service in favor of the community. With the coordination of primary and secondary transport within the regional hospital network and the national reference networks, including maternal transport (STAM) and neonatal transport (STEN), the regional coordination of transplants including the transport of organs and tissues and transplant candidates, and the coordination of the regional blood system, with particular reference to the exchange and compensation of blood and blood components [18], in 2017, the helicopter rescue service was launched in the Sardinian region and with the Performance Plan (hereinafter PDP) defined by AREUS for the three-year period of 2019—2021, which responded to the need to activate a channel of communication and direct relationship with citizens (accountability), with regard to priorities, actions and results expected by the company management. However, the PDP was also aimed at operators and all corporate human resources. In order to highlight all the components of the same administration, its strategic structure and the key variables were subject to monitoring and evaluation, in order to improve the awareness of staff with respect to the objectives of the administration. That is, it served to communicate within the "sense of direction" of the Company [19]. The key to being able to operate at its best and allow optimal cooperation between operators will always be communication and continuous reflection on what has been done to avoid the possible repetition of errors. It is therefore essential to discuss the interventions carried out and understand together what can be improved, both from the perspective of those who coordinate and from those who actually operate.

A strategy to be able to share knowledge and experiences between military and civilian helicopter rescue nurses and implement training would be the organization of periodic joint events, such as the Grifone exercise, which takes place every year in Italy and involves civilian and military organizations. The Grifone is an air Search and Rescue (SAR) exercise, which has the purpose of training the personnel of the Air Operations Command (COA)

(Desk Rescue Coordination Center/RCC), the Flight Departments, and the Authorities in charge of logistic support, planning, direction, conduct and support for search and rescue operations, in favor of aircrew victims of air accidents in mountain environments [20].

During the Grifone exercise, the differences between the management of a health emergency involving the civil and military spheres emerge, since in the civilian sphere, there is adequate availability of medical resources and the patients are located in safer areas, the phase pre-hospital is rapid or in any case has reduced times compared to the impervious/hostile environment, and the evacuation times towards the treatment sites are rapid. Concerning the military, there are limited medical resources and rescuers are often isolated, patients are located in unsafe areas, the pre-hospital phase can be protracted and therefore evacuation can be delayed, and this results in in a number of victims killed in combat near the place of injury, before the wounded person can reach the advanced medical post. The elaboration of codified intervention plans, aimed at rationalizing the use of resources for a correct response to the enormously increased needs, involves the use of three instruments: strategy which consists of planning and elaboration by levels and on the basis of risk hypotheses, linked to specific territorial conditions, which must be prepared in time and tested [21]. The intervention plans must include the problems of all the organizational aspects of the rescue: communication, public order, transport and medical rescue. The implementation of the plans must be feasible for any type of event and territory, and at any time. Another instrument is logistics, understood as the set of people and means to make the intervention plan operational and face the event, and tactical considering of the implementation of health rescue plans [22].

Thanks to this exercise and other events that may involve military and civilian personnel, the various techniques can be implemented and shared in order to have shared intervention protocols in Italy as well, which unfortunately are still missing. It could be the strategy of informing the populations of the areas at risk, identified early and developing procedures that allow the population involved to guarantee the first fundamental aid to the most serious victims waiting for qualified and effective rescue. This simple hypothesis would allow to wait for help, by eliminating the interval free from treatment [16].

Another valid proposal would be specific training starting from degree courses or even periodic courses, which should be open to a larger number of participants, including lay people, who could, thus, cooperate with the professional figures, such as doctors and nurses, and work in safety and to the best of their abilities.

**Author Contributions:** Conceptualization, F.L.; Methodology, M.R.L.; Investigation, F.L., C.M. and G.O.; Resources, C.M. and G.O.; Data curation, C.I.A.G.; Writing-original draft preparation, F.L., C.M. and M.R.L.; Writing-review and editing, F.L., C.M., M.R.L. and C.I.A.G.; Visualization, R.J.F. and I.D.A.G.; Supervision, C.I.A.G. and R.R.; Project administration, C.I.A.G., R.R. and G.O. All authors have read and agreed to the published version of the manuscript.

**Funding:** This research received no external funding.

**Institutional Review Board Statement:** Not applicable.

**Informed Consent Statement:** Informed consent was obtained from all subjects involved in the study.

**Data Availability Statement:** The datasets used and analyzed during the current study are available from the corresponding author on request.

**Conflicts of Interest:** The authors declare that they have no competing interest in this work.

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
