# Peer review of "Study on the New Role of Civil and Military Air Rescue Nurses in the Italian Context"

_nursrep, doi:10.3390/nursrep13010044_

Round 1

Reviewer 1 Report

The study on the new role of the civil military air rescue nurse in the Italian context is considered important due to the performance of nurses from critical areas departments that operate in the helicopter rescue service.

The manuscript shows the achievement of the goal by investigating the role of the civil and military helicopter nurse.

To improve the quality of the manuscript, the authors should explain in detail:

1. The causes of the low number of interviews (15) and make an analysis of the quality of the sample from the statistical approach

2. Expand the limitations of the study. Because there are no agreements.

3. Propose a strategy to share knowledge and experiences of the civil air rescue nurse and the military air rescue nurse.

Author Response

Thank you for your valuable cooperation, we have made the following changes:

  1. The causes of the low number of interviews (15) and make an analysis of the quality of the sample from the statistical approach

From  line 168 to 170

In addition, with the total number of interviews, information saturation was obtained. This made it possible to obtain fairly comprehensive information and thus be able to successfully conclude the study.

  1. Expand the limitations of the study. Because there are no agreements.

From line 509 to 515

The limits of the study are linked to the non-possibility of internship in the helicopter bases since an agreement between the University of Cagliari and the AREUS company was not active at the time of the study since it would probably have been more complete, a study carried out at 360 °. Due to its characteristics, both social and organizational as such and as a qualitative study, the results obtained cannot be extrapolated to other contexts. In addition, one of the authors of this study works as a nurse in this institution and this can influence the responses and behavior of the interviewees.

  1. Propose a strategy to share knowledge and experiences of the civil air rescue nurse and the military air rescue nurse.

From line 670 to 705

A strategy to be able to share knowledge and experiences between military and civilian helicopter rescue nurses and to be able to implement training would be the organization of periodic joint events, such as the Grifo exercise which takes place every year in Italy and involves civilian and military organizations. The Grifone is an air Search and Rescue (SAR) exercise which has the purpose of training the personnel of the Air Operations Command (COA) (Desk Rescue Coordination Center / RCC), the Flight Departments, and the Authorities in charge of logistic support, planning, direction, conduct and support for search and rescue operations in favor of aircrew victims of air accidents in mountain environments (Areus, D.G.R. 55/10 DEL 13/12/2017). During the Grifone exercise, the differences between the management of a health emergency involving the civil and military spheres emerge, since in the civilian sphere there is adequate availability of medical resources, the patients are located in safer areas, the phase pre-hospital is rapid or in any case has reduced times compared to the impervious/hostile environment, the evacuation times towards the treatment sites are rapid. Concerning the military, there are limited medical resources and rescuers are often isolated, patients are located in unsafe areas, the pre-hospital phase can be protracted and therefore evacuation can be delayed and this results in in a number of killed in combat near the place of injury, before the wounded person can reach the advanced medical post. The elaboration of codified intervention plans, aimed at rationalizing the use of resources for a correct response to the enormously increased needs, involves the use of three instruments: Strategy which consists of planning and elaboration by levels and on the basis of risk hypotheses linked to specific territorial conditions, which must be prepared in time and tested (Areonatica- Home page). The intervention plans must include the problems of all the organizational aspects of the rescue: communication, public order, transport and medical rescue. The implementation of the plans must be feasible for any type of event and territory, and at any time. Logistics, understood as the set of people and means to make the intervention plan operational and face the event. Tactical considering the implementation of health rescue plans (Area-c54.it). Thanks to this exercise and to other events that may involve military and civilian personnel, the various techniques can be implemented and shared in order to arrive one day to have shared intervention protocols in Italy as well, which unfortunately are still missing. It could be the strategy of informing the populations of the areas at risk, identified early and developing procedures that allow the population involved to guarantee the first fundamental aid to the most serious victims waiting for qualified and effective rescue. This simple hypothesis would allow to wait for help by eliminating the interval free from treatment, although these will be very simple. (Sutherland et al., 2021)

Reviewer 2 Report

The paper entitled "Study on the new role of the civil and military air rescue nurse in the Italian context" provides an Italian case study based on the Sardinia region in particular.  The main objective was to explore the operational context of emergency medical nurses, understanding how they work outside the department of origin, and how their training has affected the possibility of establishing themselves outside it and thus being able  to become part of a context that is considered to be of the highest level.  

But the author needs to pay attention to configuring tables and figures within the paper. what is tagged as Figure 1. Partecipant. is actually a Table. and there are several spelling errors here and there. The reference section is too bad. Please followed some citation styles such as APA or Chicago. Please spend more time rewriting and proofreading tasks. Thank you very much. 

Author Response

Thank you for your valuable cooperation, we have made the following changes:

Line 216:

Table 1. Home base, department of origin and gender of partecipants

We used citations with APA, and, we have also corrected the form and content of the article as you suggested.

Reviewer 3 Report

First of all, I want to congratulate the authors of this study. In fact, it is a relevant topic for the academic and scientific community.

I give some observations, which should be taken into account by the authors.

Introduction – improve the theoretical framework and the object of study of the investigation.

Materials and Methods – the use of resources such as the Consolidated criteria for reporting qualitative research (COREQ): a 32-item checklist for interviews and focus groups is not described. What is the reason? This aspect is essencial, according the methodological criteria.

Line 135 – Word “Partecipant”

Topic 4 – Does not exist more limitations of the study?

Globally, the study is little referenced and with few recent studies. Authors are recommended to correct this aspect.

Author Response

Thank you for your comments, below are the corrections made to the article based on your suggestions:

  • From line 47 to 57

Introdution:

With the aim of guaranteeing, managing and making homogeneous, in the territory of the Region, the regional emergency and urgency of Sardinia (AREUS) is established with regional law 17 November 2014, n. 23. The Company is responsible for carrying out the tasks related to the emergency-urgency currently carried out by the 118 operations centers at the health companies, including the helicopter rescue service, as well as the coordination functions in the transport of people, including newborns, organs and tissues, exchange and compensation of blood and blood components, and ensures coordination with all the companies of the regional health system and the institutions involved in order to guarantee  the effectiveness of the emergency and urgency health response. (Sardegna, 2018).

  • Materials and Methods:

We apologize for not being explicit with the 32 items of the qualitative research using the interwiew as an instrument, we consider that it was implicit in a certain way in the article but we accepted the suggestion, and we decided to describe in the paper.

From line 119 to 159

We carried out the study with interviews conducted by FL, a female student of the nursing degree course, with previous work experience in the emergency- urgency because he has been part of a medical aid association for 7 years. The relationship with the interviewees began on the same day as the interviews, as the researcher did not know the interviewees, but they were informed during the beginning of the interview about their personal goals. There was also a personal motivation for conducting the aforementioned study. The method was with a phenomenological orientation and in order to implement this, the 40 nurses operating in the Sardinia region's helicopter rescue service were selected, 15 of whom decided in total freedom to join the project, thus becoming the champion of the study. Those who decided not to participate did so either due to lack of time. Participants were invited via email and the interviews were conducted via video call. During the interviews, no other people outside the study were present. The participants all came from critical area departments or from the local emergency service and were mostly male. All respondents had considerable previous work experience. The questions have been designed to be compatible with the chosen topic, in full respect of ethics and privacy. Questions selected for interviews were relevant to the research, focus on a topic, with brevity and clarity and appropriate language for the respondents; furthermore, the delicate topics were not addressed directly so as not to make the participants uncomfortable and above all no double-answered questions were formulated and space-time contexts were delimited. The pilot interview was submitted to 2 helicopter rescue nurses operating in the Sardinia region. They did not suggest any changes to the questions and they were clear and compressed. All this was structured in such a way as to understand how the respondents spoke of the investigated phenomenon allowing them to respond with a natural language and thus have immediate answers.

 All the interviewees were asked the same questions and data collection was to record the interviews in order to have the necessary material to work satisfactorily on the study and to avoid errors. The total duration of the interviews is 10:40 hours, after which the saturation of the data obtained with respect to the questions was evaluated and, having reached the set target, the data was encoded through revision by FL, MRL, CM and CIAG with the drafting of a coding tree and with the derivation of the themes that emerged from the data obtained with the interviews, identifying the main thematic areas and extrapolating the answers of each participant in the interview for each question proposed. Not all participants wanted to review their interviews.

3) Line 216:

Table 1. Home base, department of origin and gender of partecipants

4)From line 509 to 516

The limits of the study are linked to the non-possibility of internship in the helicopter bases since an agreement between the University of Cagliari and the AREUS company was not active at the time of the study since it would probably have been more complete, a study carried out at 360 °. Due to its characteristics, both social and organizational as such and as a qualitative study, the results obtained cannot be extrapolated to other contexts. In addition, one of the authors of this study works as a nurse in this institution and this can influence the responses and behavior of the interviewees.

Round 2

Reviewer 3 Report

It is considered that the authors took into account the suggestions for improvement.